# Production of Encecalin in Cell Cultures and Hairy Roots of *Helianthella quinquenervis* (Hook.) A. Gray

**DOI:** 10.3390/molecules25143231

**Published:** 2020-07-15

**Authors:** J. Mabel Hernández-Altamirano, Irene F. Ugidos, Javier Palazón, Mercedes Bonfill, Penélope García-Angulo, Jesús Álvarez, José L. Acebes, Robert Bye, Antonio Encina

**Affiliations:** 1Área de Fisiología Vegetal, Departamento de Ingeniería y Ciencias Agrarias, Universidad de León, 24071 León, Spain; mabel.hdez@gmail.com (J.M.H.-A.); iferu@unileon.es (I.F.U.); penelope.garcia@unileon.es (P.G.-A.); jmalvf@unileon.es (J.Á.); a.encina@unileon.es (A.E.); 2Laboratorio de Cultivo de Tejidos Vegetales, Jardín Botánico, Instituto de Biología, Universidad Nacional Autónoma de México, Ciudad Universitaria, Mexico City 04510, Mexico; 3Unitat de Fisiologia Vegetal, Facultat de Farmàcia, Universitat de Barcelona, Avda Joan XXIII 27-31, 08028 Barcelona, Spain; javierpalazon@ub.edu (J.P.); mbonfill@ub.edu (M.B.); 4Laboratorio de Etnobotánica, Jardín Botánico, Instituto de Biología, Universidad Nacional Autónoma de México, Ciudad Universitaria, Mexico City 04510, Mexico; bye.robert@gmail.com

**Keywords:** medicinal plant, encecalin, *Helianthella quinquenervis*, *Agrobacterium rhizogenes* A4, chromenes, sonication, secondary metabolites

## Abstract

Plant cell and organ cultures of *Helianthella quinquenervis*, a medicinal plant whose roots are used by the Tarahumara Indians of Chihuahua, Mexico, to relieve several ailments, were established to identify and quantify some chromenes with biological activity, such as encecalin, and to evaluate their potential for biotechnological production. Gas chromatography–mass spectrometry (GC-MS) analysis corroborated the presence of quantifiable amounts of encecalin in *H. quinquenervis* cell cultures (callus and cell suspensions). In addition, hairy roots were obtained through three transformation protocols (prick, 45-s sonication and co-culture), using wild type *Agrobacterium rhizogenes* A4. After three months, cocultivation achieved the highest percentage of transformation (66%), and a comparable production (FW) of encecalin (110 μg/g) than the sonication assay (120 μg/g), both giving far higher yields than the prick assay (19 μg/g). Stable integration of *rolC* and *aux1* genes in the transformed roots was confirmed by polymerase chain reaction (PCR). Hairy roots from cocultivation (six months-old) accumulated as much as 1086 μg/g (FW) of encecalin, over three times higher than the cell suspension cultures. The production of encecalin varied with growth kinetics, being higher at the stationary phase. This is the first report of encecalin production in hairy roots of *H. quinquenervis*, demonstrating the potential for a future biotechnological production of chromenes.

## 1. Introduction

*Helianthella quinquenervis* (Hook.) A. Gray (Asteraceae) is a perennial herb that grows in montane and subalpine coniferous forest of the Rocky Mountains of the western United States of America and Canada where it is known as the aspen sunflower, and in pine-oak forests of the Sierra Madre of Chihuahua and Nuevo Leon, Mexico [1] (Figure 1). The Tarahumara Indians of Chihuahua know it as rarésoa [2] or corsonero (probably derived from escorzonera) and employ it medicinally. Dry and powdered roots are used to alleviate coughs and muscle aches and to treat gastrointestinal diseases and ulcers [3]. Crude extracts of *H. quinquenervis* roots have shown antimicrobial activity against *Trichophyton mentagrophytes*, one of the fungi that cause ringworm infections in humans [4], and antiprotozoal activity [5].

Phytochemical analysis of the roots and aerial parts of *H. quinquenervis* isolated and identified different chromenes (benzopyrans) such as encecalin and demethylencecalin, and benzofurans such as euparin (Figure 1) [6,7,8]. Diverse bioassays have been used to verify the biological activity of some of the 200 chromenes and benzofurans identified in *H. quinquenervis* and other plant species, most of them belonging to the family Asteraceae [9]. Many of these secondary metabolites have insecticidal and cytotoxic properties [10,11]. Encecalin and demethylencecalin inhibit the radical growth of seedlings of several crop plants and weed, including *Amaranthus hypochondriacus* and *Echinochloa crusgalli* [6,12,13,14]. In isolated spinach chloroplasts, they inhibit ATP synthesis, proton uptake and electron flow and disturb photosystem II. On the other hand, euparin inhibits ATP synthesis, proton uptake and basal and phosphorylating electron transport, suggesting that euparin suppresses phosphorylation in chloroplasts, acting as an energy-transfer inhibitor [3]. Additionally, encecalin and demethylencecalin exhibit marginal cytotoxicity against several human tumour cell lines [6]. Encecalin has also been used as a key intermediate for the synthesis of diverse ethers, amides and amines that were checked for antiprotozoal activity [15].

The source of these compounds continues to be wild medicinal plants with ethnobotanical value. However, the commercial production of secondary metabolites from wild plants is hampered by fluctuating environmental conditions and low yields. In addition, harvesting the medicinal root kills the plant, thus preventing seed production, which is critical for regeneration and maintenance of natural populations. Consequently, it is important to develop alternative systems of production of medicinal plants, especially those with limited ecological and geographical ranges, in case the demand for these plants suddenly overrides the local supply and drives the wild populations into extinction. Such alternative methodology should be appropriate for transfer to institutions located in the region of the source plant material so that inhabitants can benefit from the biotechnological applications. An alternative approach to secondary metabolite production is organic synthesis, but this is frequently constrained by the complexity of the target structure [16].

More sustainable strategies to obtain plant secondary metabolites of interest for the agrochemical, pharmaceutical and food industries involve in vitro culture methodologies, together with the use of modified tissues, organs and whole plants [17,18,19]. Transformed hairy roots cultures, obtained by the infection of plant cells with *Agrobacterium rhizogenes*, are an option for the biotechnological exploitation of metabolites due to their high growth rate, genetic stability and capacity for growing in hormone-free media. In addition, this biotechnological strategy allows the production and/or biotransformation of highly diverse molecules [20,21]. *A. rhizogenes* is a gram-negative soil bacterium that can cause the hairy root syndrome in many plants. Infections in wound sites are followed by the transfer, integration and expression of T-DNA from the root-inducing (Ri) plasmid and the subsequent development of the hairy root phenotype [22]. The hairy roots, once separated from the tissue of origin, can continue to grow in hormone-free media. The *Agrobacterium* genes implicated in the rhizogenesis, known as *rol* and *aux* genes, are located in two regions which are independently transferred to the plant cell [23]: the left transferred (T_L_-DNA) and right transferred (T_R_-DNA) regions, respectively. The *aux* genes are involved in auxin biosynthesis and/or sensitivity. In the last decade, the use of hairy root culture technologies for pharmaceutical production has progressed from laboratory-scale to an industrial level [24]. The secondary metabolite yields of transformed root cultures can be comparable with those of intact plants, and in some species may be increased by elicitation [25,26].

Although encecalin and related compounds are relatively simple molecules, little is known about their production in in vitro culture systems. Furthermore, the biological importance of these chromenes and benzofurans, calls for new production strategies with potential to be optimized for their large-scale and cost-effective production. In this work, callus, cell suspension and hairy-root cultures of *H. quinquenervis* were obtained in order to evaluate the production of encecalin and other related secondary metabolites with potential value in the agricultural and pharmaceutical area.

## 2. Results and Discussion

### 2.1. Germination Treatments and Callus Induction

Mechanical scarification (Figure 2a) was a precondition for the germination of *H. quinquenervis* seeds, which present an extra-embryonic dormancy due to the hardness and poor permeability of the pericarp. Scarification under two different conditions (in vitro and soil) clearly affected the germination rate, increasing it from 13% and 38%, respectively, for unscarified seeds to 90% for scarified seeds after thirteen days (data not shown).

A similar result was reported in seeds of *Anthemis chrysantha* (Asteraceae); after breaking seed dormancy by mechanical scarification or excising the embryo, 71% germination was achieved [27]. These results indicated that the presence of the pericarp and testa inhibits seed germination.

*H. quinquenervis* scarified seeds were germinated in vitro and cultured in axenic conditions. Two months after germination, plants were 5 to 8 cm in height showing balanced stem and leaf development (Figure 2b). These stems and leaf were selected as explants (Figure 2c) for callus induction.

After testing both types of explants with different combinations and ratios of auxin/cytokinin, the selected medium was 6-benzylaminopurine (BAP, 1 mg/L) + 2,4-dichlorophenoxyacetic acid (2,4-d) (0.1 mg/L). Leaf explants were chosen for their high percentage of callogenesis (90%) and low oxidation rate (data not shown). The biomass produced by callus from leaf explants was 0.273 mg FW/explant, showing a friable consistency and pale-yellow color. The treatment with BAP (1 mg/L) + naphthaleneacetic acid (NAA, 2 mg/L) on leaf explants yielded a higher quantity of callus biomass (0.318 mg FW/explant), but it was discarded because of an oxidation process that appeared during subculturing, and the compact consistency of the calli.

When stem explants were exposed to combinations of auxin/cytokinin, they generated 58% of calli in the presence of BAP (1 mg/L) + NAA (2 mg/L) and 60% with BAP (1 mg/L) + 2,4-d (0.1 mg/L). Because both combinations generated less biomass than in the case of leaf explant and resulted in oxidation and in the production of less friable calli, the stem explants were discarded for future experiments.

Once the optimum conditions for callus induction were established, one set of calli was kept in the dark (Figure 2d) and another set under a photoperiod of 16:8 (Figure 2e). The calli cultured under light were greenish and friable, with some compact and hard areas. On the other hand, those maintained in the dark were a pale-yellow color and more friable. For this reason, dark cultured calli were chosen for obtaining cell suspension cultures.

### 2.2. Characterization of Cell Suspension Cultures

Cell suspension cultures (Figure 2f) were derived from leaf calli growing in MS medium with BAP (1 mg/L) + 2,4-d (0.1 mg/L) in the dark. To achieve these, calli were transferred to a liquid medium of the same composition and allowed to grow on a rotary shaker in the dark. Figure 3 shows the growth kinetics of the suspension-cultured cells. After a lag phase of two days, the biomass dry weight (DW) progressively increased, reaching the stationary phase at day 8. Once the cell suspension cultures grew stable—after more than 8 subcultures—a 70% viability was determined during active growth.

*H. quinquenervis* suspension cultured cells grew forming aggregates, which seemed to function as generators of new cells. Filtration through different sized meshes resulted in diversely sized cell aggregates in the stationary phase (Figure 4a), ranging from large (greater than 2 mm and accounting for over 40% of the DW) to very small (between 120 and 400 µm and accounting for 30% of the DW), as well as individual cells.

Although cell suspension cultures of *H. quinquenervis* presented great variation in the size of cellular aggregates, this did not affect the culture viability. In samples obtained by differential filtration, large cell aggregates with high viability were observed (Figure 4d). These appeared to be the generators of cells that would be part of a suspension with cells of more homogeneous size.

The establishment and characterization of suspension cultures to produce compounds of interest has been reported in cultures of *Stevia rebaudiana* in shake flasks, as a strategy to obtain an in vitro cell line producing stevioside [28]. In the case of medicinal plants, many strategies have been developed to induce high-yield cultures [29], including elicitation to stimulate production [30] and the transformation of metabolic pathways [31]. With this taken into account, it is believed that the establishment of suspension cultures of *H. quinquenervis* would allow the study of chromenes.

### 2.3. Hairy Root Induction

The leaf explants, taken from in vitro-germinated plants, transformed by *A. rhizogenes* strain A4 and maintained in hormone-free solid medium with cefotaxime, began to develop small roots after twelve days of transformation, sprouting from the wound sites in all three of the transformation assays (Figure 5a). After 20 days (Figure 5b), the length and number of the roots had increased. The roots were pale yellow, and callus formation at the base of the roots was observed after 30 days (Figure 5c), but root growth was not affected. After several subcultures in 50% liquid MS medium with antibiotics, the roots were subcultured in liquid MS medium without antibiotics, in which they continued to grow. The roots showed typical transformed root morphology: plagiotropic growth, thickness and the presence of root hairs (Figure 5d).

The susceptibility of *H. quinquenervis* plants to transformation by *A. rhizogenes* was tested using three techniques. Besides the commonly used methods of pricking and co-cultivation, sonication was also tried in which the wounded explants were exposed to ultrasound [32]. The results of the three different assays (prick, sonication and co-culture) showed that *H. quinquenervis* was susceptible to transformation by *A. rhizogenes* strain A4 (Table 1). After forty days, the lowest response to the transformation was observed in the prick assay (17%), in the number of infected explants as well as root-generating explants. Moreover, the infected explants showed oxidation over time. In the sonication assay, the transformation percentage was higher (40%), and numerous hairy roots (56) were obtained. In the same way, the sonication technique was efficiently used to obtain hairy roots from explants of *Verbascum xanthophoeniceum*, with a 75% transformation rate [32]. However, in the *H. quinquenervis* explants, the co-culture gave the highest percentage of transformation (66%) of the three assays. Although the number of roots per explant varied among the assays, no differences were observed in the morphology of roots, which were generally thin and pale yellow. In all three experiments, the remainder of the explants were necrosed or reinfected (i.e., subsequently infected by *A. rhizogenes*), which prevented their subsequent use. Those transformed roots in which the infection by *A. rhizogenes* could not be eliminated were discarded.

After forty days, several hairy roots from each assay were selected and maintained in liquid MS medium with sucrose and the antibiotic. After the fourth subculture under these conditions, the eventual presence of *Agrobacterium* and the transformation of the cultures were analyzed by PCR. At ninety days after transformation, a sufficient number of roots of increased length (Figure 5d) were obtained for the quantification analyses.

The success of transformation depends on several factors, such as the type of explant, its age and the species, as well as the *Agrobacterium* strain [33], since some are more virulent than others. The transformation capacity is partially related to the variety of Ri (root inducing) plasmids within each bacterial strain. In a previous trial, poor results were obtained with the *A. rhizogenes* LBA9402 strain (data not shown). In contrast, the A4 strain successfully produced hairy roots. Transformation efficiency in *Fagopyrum tataricum* with the same strain was 100% [34], being somewhat lower in the *H. quinquenervis* explants, at 66.7%.

### 2.4. PCR for the Confirmation of Root Transformation

The transformation of *H. quinquenervis* hairy roots was confirmed by PCR. Amplification of *rolC*, *aux1* and *virG* primers showed that the transformation system was complete and stable (Figure 6). The PCR results revealed that T_L_-DNA (*rolC*) and T_R_-DNA (*aux1*) fragments were successfully inserted in the three assays (prick, sonication and co-culture). Successful elimination of *A. rhizogenes* infection in the three cultures was confirmed by the absence of the band representing the *virG* amplified product, which was only observed in the sample corresponding to the *A. rhizogenes* A4 strain. The absence of infection by *A. rhizogenes* was also confirmed in leaves (used as explants), as well as in *H. quinquenervis* untransformed roots.

The complete and stable transformation of *H. quinquenervis* hairy roots was confirmed by identifying the bacterial genes *rolC* and *aux1* using PCR analysis. Likewise, contamination by *A. rhizogenes* was discarded in both nontransformed and transformed roots, when no band was observed for the *virG* gene. The amplification of *rolC* (534 bp) in the roots produced by the different techniques indicated that the T_L_-DNA fragment was integrated into the plant genome, which is essential for hairy root induction [32]. Amplification of *aux1* confirmed that the T_R_-DNA was also integrated. The segment, containing *aux1*, *aux2* and *ags* genes, controls opine and auxin biosynthesis [35], the latter being indispensable for the growth and differentiation of the roots, which did not need auxins in the culture medium. The complete integration of T_R_-DNA and T_L_-DNA was confirmed in the *H. quinquenervis* transformed roots, but not in the nontransformed roots of *L. album* (L-), used as a negative control, nor in the nontransformed roots of *H. quinquenervis*, as expected. The confirmed presence of the *rol* and *aux* genes in the transformed roots of *H. quinquenervis* is valuable for the role they would play in the synthesis of secondary or specialized metabolites. Such is the case of *Artemisia carvifolia* and its role in enhancing the production of antioxidants [36]. This relationship has been well documented in a wide number of plant species [37].

### 2.5. Detection and Quantification of Chromenes In Vitro Cultured Cells of H. quinquenervis

In its wild form, *H. quinquenervis* produces chromenes and benzofuranes, which have been identified in aerial parts and roots [7]. In order to determine if *H. quinquenervis* in vitro cultures produce secondary metabolites of interest, an assay was developed to produce calli in sufficient quantities for phytochemical analysis. A frequent drawback of undifferentiated cell masses is that they do not produce significant amounts of target compounds, as secondary metabolism usually relies on cell differentiation [38]. However, secondary metabolite production has also been described in undifferentiated cells of other Asteraceae species, such as *Piqueria trinervia*, whose calli produced piquerol A, a monoterpene with antifungal and allelopathic activity [39], and *Rudbeckia hirta*, whose calli and cell suspension cultures produced pulchelin E, a sesquiterpene lactone with immunostimulating properties [40].

GC-MS analysis of chloroform extracts from *H. quinquenervis* calli revealed the presence of encecalin (Table 2 and Figure 7).

Because the synthesis pathways of chromenes and benzofuranes are not well determined, and there are compounds such as terpenes that depend on the incidence of light for their synthesis [41], it was interesting to assess whether light affected synthesis in any way. In this respect, it was found that lighting was not a prerequisite for encecalin biosynthesis as both light and dark-grown calli produced encecalin (Table 2). In addition, encecalin production by calli revealed that calli kept in the dark produced 1.7 times more encecalin than those under photoperiodic conditions (Table 2).

Encecalin was also detected in suspension-cultured cells of *H. quinquenervis* (Table 2) that produced almost twice (1.98 times) as much of encecalin than calli kept in a photoperiod (Table 2). The higher amount of encecalin produced by cell suspension cultures may be explained by the stress provoked by the immersion and shaking of the cells, which could elicit their synthesis [42].

### 2.6. Hairy Root Growth and Encecalin Quantitative Analysis

GC-MS analysis of chloroform extracts from two-month-old roots from in vitro plants revealed the presence of some of the most important secondary metabolites of *H. quinquenervis,* such as encecalin and the structurally related compound euparin (Figure 1), although in low concentrations. In the three transformed root lines obtained by prick, sonication and co-culture techniques, after three months encecalin was produced in sufficient quantities to perform quantification (Figure 8).

Encecalin production significantly varied in relation to the method used for *H. quinquenervis* transformation as determined by one-way ANOVA (F(2,6) = 457.9; *p* < 0.0001). Although co-culture resulted in the highest transformation frequency, it provided roots with a slightly lower concentration of encecalin (110 ± 5 μg/g FW) than the sonication assay (120 ± 6 μg/g FW); whereas the prick assay resulted in a low transformation frequency and an encecalin concentration of 19 ± 1 μg/g FW.

Even though the production of encecalin was highest when using the sonication technique, the fact that the percentage of infection was much higher for the co-culture technique, and that the encecalin production difference, although significant, was only slightly higher for the sonication method, the co-culture method of transformation was chosen to continue analyzing the production of encecalin for a longer period. After six months, a concentration of 1086 ± 31 µg/g FW was achieved (Table 2). This result confirmed that in *H. quinquenervis*, the synthesis of encecalin was more abundant in organized structures (roots) than in disorganized tissues such as calli or cell suspension; also, the higher the level of differentiation of roots, the greater the production of encecalin.

Based on the transformation percentage and accumulation of encecalin, sonication and co-culture were the best methods to develop encecalin-producing hairy roots, probably due to morphology and the amount of root biomass generated. Although several DNA uptake methods are available to produce hairy roots, the results can be random and unpredictable [43]. In our case, sonication enhanced the number of roots produced by the explant, as well as transformation frequency and encecalin production, compared with the pricking method. Sonication improved the efficiency of *Agrobacterium* transformation through numerous microwounds on the surface and deep within the target tissue [44]. The technique has been successfully used in several species and types of tissues and organs: immature cotyledons of soybean [44], flax hypocotyls and cotyledons [45], nodal explants of *Withania somnifera*, which is important for its therapeutic phytocompounds, and *Paspalum vaginatum* calli [46].

Regarding the growth of hairy roots, they reached the exponential phase on the 16th day and the stationary phase on the 26th day of the culture period, producing the largest biomass increment on the 28th day (Figure 9a). The contents of encecalin in 6-month-old hairy roots per unit of DW (Figure 9a), were inversely correlated with root growth: as the roots continued to grow, the amount of encecalin diminished. Correspondingly, when root growth began to decline on day 30, the amount of encecalin increased considerably, attaining the highest production (60 µg/g DW) at day 40. Oxidation of the culture medium was observed matching the decrease of roots. When analyzed per explant, encecalin levels were found to increase monotonically, with a sharp acceleration at around day 30 (Figure 9b). Similar studies using *Pueraria candollei* hairy roots reported a growth behavior that of *H. quinquenervis*, finding a higher content of the compounds of interest (e 8-C-glucoside of daidzein) after the stationary phase [47].

The higher production of encecalin in the stationary phase may be due to several factors, including cell differentiation bound to the reduction in culture medium nutrients, and sufficient accumulation of intermediates for secondary metabolism. The observed trend was similar to that of other biotechnology culture systems producing secondary compounds, which accumulated after the exponential growth phase when the cells entered the stationary phase; for example, betalain production in hairy roots of *Beta vulgaris* [48], and paclitaxel in cell cultures of *Taxus baccata* [49]. In general, high rates of biomass production have negative effects on secondary metabolite yields [50,51].

The presence of encecalin was confirmed in young leaves and roots by GC-MS. In contrast, there are no reports of demethylencecalin in young leaves, nor in roots, of in vitro plants or hairy roots, whereas it has been found in aerial parts [6,7]. This indicates that the synthesis of demethylencecalin is at least partially related to plant ontogeny. The differential production of these secondary metabolites during ontogeny could be related to resistance to pathogen attack [52], as encecalin is known to have an insecticidal effect [9,10]. A similar response has been reported in leaves of seedlings of *Ageratina adenophora* in which chromenes (encecalin and demethylencecalin) may be the most important compounds in chemical defense, whereas in older plants sesquiterpene and chlorogenic acid reportedly take over as protective chemicals [53].

The diverse in vitro plant culture techniques provided the opportunity to amplify the biotechnological potential of *H. quinquenervis* to produce encecalin in calli, cell suspensions and hairy roots. This is the first report regarding encecalin by using *H. quinquenervis* in vitro culture. Should the Madrean populations of *H. quinquenervis* be subjected to exploitation, practical biotechnological alternatives coupled with conventional harvesting can be applied in the region to ensure the sustainable management of this biocultural resource.

## 3. Materials and Methods

### 3.1. Seed Germination and Plant In Vitro Growth

Cypselae (achene-like fruits) of *Helianthella quinquenervis* were collected initially from wild populations in secondary pine-oak forests of the municipio de Bocoyna, Chihuahua, Mexico (Figure 10). A corroborative voucher specimen was deposited at the National Herbarium (Bye 39646; MEXU 1517385) of the Institute of Biology, UNAM, Mexico. The national permit for the collection of plant material was issued by Mexico’s Subsecretaría de Gestión para la Protección Ambiental under the certificate SGPA/DGGFS/712/2658/18.

Seeds were scarified by removal of the pericarp and the seed coat, disinfected with 70% (*v*/*v*) ethanol for 2 min, 10% (*v*/*v*) H_2_O_2_ for 15 min, and 4% NaOCl for 20 min, and then rinsed three times with distilled water. Scarified seeds were transferred for germination to glass flasks containing Murashige and Skoog medium (MS) [54] with vitamins, supplemented with 30 g/L sucrose and solidified with 8 g/L agar. The pH was adjusted to 5.7 before autoclaving for 15 min at 121 °C. The in vitro grown seedlings were kept under a photoperiod of 16 h light/8 h darkness at 25 °C for two months.

### 3.2. Calli and Cell Suspension Cultures

Calli were obtained from leaves and nodal segments of 1 cm^2^, which were placed on MS medium supplemented with BAP, 1 mg/L, together with NAA, 2 mg/L or 0.1 mg/L, incubated at 25 °C and subcultured every 25 d on fresh solid medium. Both combinations were used to induce callogenesis, but only the latter was chosen to maintain the calli. At this point the calli were soft and friable, which was ideal to obtain cell suspension cultures.

To assess the effect of photoperiod upon callus induction and growth, one set of cultures was kept in darkness and another in 16 h light/8 h darkness.

Friable callus pieces (2 g) were placed in 50 mL liquid MS medium in 250 mL flasks containing 1 mg/L BAP and 0.1 mg/L 2,4-d. Three flasks were used to obtain the cell suspension cultures. After ten days, the cultures were filtered and transferred to fresh medium. In this process, oxidized material was removed, and after three subcultures the contents of the flasks were mixed and filtered through a mesh (400 μm) to obtain an homogeneous suspension culture.

The flasks were shaken at 120 rpm on an orbital shaker, kept in darkness for two months and thereafter under a photoperiod of 16 h light/8 h darkness at 25 °C. The cell suspension was cultured for 12 d to estimate growth curve kinetics. Cell clusters were measured by differential filtration using nylon meshes of various pore size (3000 μm to 22 μm), and the result was expressed as the percentage of the dry weight (DW) retained by each filter.

Viability of the suspension culture was observed by the fluorescein diacetate method as described by Duncan and Widholm [55]. One hundred μL of 0.5% (*w*/*v*) fluorescein diacetate stock solution in acetone was diluted in 0.9 mL of water. The resulting working solution was mixed 1:1 (*v*/*v*) with the cell suspension, kept for 2 min in the dark and then the bright green fluorescence emitted by viable cells was analysed on a Nikon epifluorescence microscope equipped with a Nikon UV-2A filter (330–380 nm excitation, 400 nm dichroic mirror and 435 nm barrier filter) (Nikon Instruments Inc., Melville, NY, USA). Five optical fields with nearly a hundred cells in total were counted, and the viability was expressed as percentage of viable cells compared to total cells.

### 3.3. Hairy Roots Induction

A preliminary study of transformation and induction of *H. quinquenervis* hairy roots was carried out. Explants were infected with *A. rhizogenes* strain A4, which was grown in solid YEB medium containing 10 g/L yeast extract, 10 g/L pancreatic peptone, 5 g/L NaCl and 15 g/L agar, pH 7.

Before infection, the *A. rhizogenes* strain was grown in liquid YEB medium, shaken at 250 rpm and incubated at 28 °C for 12 h. After 1:1 dilution, it was incubated again for 1 h. To check bacterial growth, optical density (O.D.) was measured at 600 nm and when the O.D. was between 0.8 and 1.0, the culture was centrifuged at 4000 rpm for 20 min. The pellet was resuspended in MS medium for the infection.

Three protocols were used for transformation: pricking, sonication and co-culture. In all cases sterile leaves sections of ~1 cm^2^ from in vitro plants were used. Each protocol was based on the use of five plates, each with six or nine leaf sections (i.e., 30–45 explants). Instead of using acetosyringone as an infection stimulator, the explants were left to stand for two hours prior to infection.

Infection by pricking was carried out with a sterilized needle impregnated with *A. rhizogenes* from colonies of bacteria grown in YEP medium for 48 h. Inoculation was done through incisions in the midrib of the lower leaf surface then the explants were incubated in MS medium without antibiotics or hormones in darkness for 48 h at 25 °C. Inoculations without bacteria were performed in the same way as controls and it was found that the explants did not form roots.

The sonication protocol (also called sonication-assisted *Agrobacterium*-mediated transformation (SAAT)) was based on that of Georgiev et al. [32]. The explants were placed in tubes with 20 mL of MS medium and *A. rhizogenes* culture 1:1, and then exposed to ultrasound at 35 kHz for 45 s. Excess medium was removed and explants were transferred to solid MS medium without antibiotics or hormones, and maintained in darkness for 48 h at 25 °C.

In co-culture, 20 mL of *A. rhizogenes* culture was centrifuged at 4000 rpm for 10 min. The supernatant was removed and the pellet was resuspended with 10 mL of MS medium without antibiotics or hormones and placed in a Petri dish (containing 10 mL of MS medium without bacterial culture). The explants were immersed in this suspension for 20 min. The co-culture medium was removed, the explants were dried with sterilized wipes, transferred to solid MS medium without antibiotics and maintained in darkness for 48 h at 25 °C. Explants not co-cultured with the bacteria did not show any organogenic response.

At 40 days, all of the explants survived and most of them formed different numbers of roots, which ranged from 1 to 8.

### 3.4. Use of PCR for Transformation Analysis

Genomic DNA of hairy roots was isolated using the protocol described by Dellaporta et al. [56]. PCR was performed using PuReTaq Ready-To-Go™ (GE Healthcare, Buckinghamshire, UK). The thermocycler (MJ Research Inc., Watertown, MA, USA) was a Mini Cycler™, MJ Research (35 cycles). The gene-specific primers used to confirm the transformations are specified in Table 3.

The amplified segments were separated in 1.5% agarose gel and stained with RedSafe™. The genomic DNA of *A. rhizogenes* was used as a positive control to detect transformed hairy roots. DNA isolation was performed by previously lysing the bacteria in boiling water.

### 3.5. Establishment of Hairy Roots in Liquid Medium and Growth Analysis

After 48 h in darkness in MS medium without antibiotics, as mentioned before, infected explants from each protocol were grown in a solid medium with cefotaxime. When the hairy roots appeared and reached a length of 4 cm, they were cut and transferred (80 mg) to 50 mL of liquid MS medium with 500 mg/L cefotaxime. They were subcultured on new medium every 10 days for 4 weeks and then transferred to liquid MS medium without cefotaxime. The liquid cultures were maintained at 25 °C in the dark and shaken at 120 rpm. To determine growth, 1 g of fresh hairy roots were inoculated in Erlenmeyer flasks with 100 mL of liquid MS medium and cultured for 40 days. After separating the hairy roots from the culture by filtration, their FW and DW were determined.

### 3.6. Detection and Quantification of Encecalin

Samples of leaves and roots from two-month-old in vitro plants, three-month-old calli, cell suspensions, as well as three-and six-month hairy roots were analyzed for the presence of encecalin. Each sample was ground with liquid nitrogen in a mortar, extracted with chloroform and stirred on a magnetic stirrer for 8–12 h. The solvent was eliminated in a rotary evaporator (Heidolph Instruments GmbH & CO, Schwabach, Germany) and the extracts were analyzed.

The analytical method for all the samples was gas chromatography coupled to mass spectrometry (GC-MS). A Varian chromatographer (Agilent, Santa Clara, CA, USA) was used, model CP-3800, with injector model 1079, and a Varian Saturn 2200 as the mass detector: VF-5 ms column, 0.25 mm in diameter and 30 m long (from Agilent), and gas helium as the carrier. The injected samples were of 1 µL, in triplicate.

Encecalin was detected by means of mass-masses with electronic ionization, using mass-mass 217 as a precursor ion and dissociated under the nonresonant excitation method. Encecalin was quantified using a fragment ion of m/z 185. The standard encecalin used (ref. ALX-350-393-M001) was manufactured and marketed by Enzo Life Sciences (Farmingdale, New York, USA).

### 3.7. Statistical Analysis

Where appropriate, data were compared by using one-way ANOVA followed by Tukey test (*p* < 0.05).

## 4. Conclusions

The optimal conditions for germination and the induction of calli and cell suspensions of *H. quinquenervis* were established in a series of experiments.

Wild plants of *H. quinquenervis* proved to be susceptible to infection by *A. rhizogenes* strain A4 using three protocols: prick, sonication and co-culture. Several important chromenes were identified in diverse in vitro cultures: encecalin in both hairy roots and callus cultures, and euparin only in the former. This is the first report of hairy root cultures in *H. quinquenervis*.

As regards the quantified production of encecalin in diverse cultures and tissues (callus, cell suspension and hairy roots), a clear advantage was observed in hairy roots, with over three times the concentration of the second best (cell suspension). This result clearly suggests the potential offered by transformed roots in developing improved production methods for biotechnological production of encecalin.

## Figures and Tables

**Figure 1 molecules-25-03231-f001:**
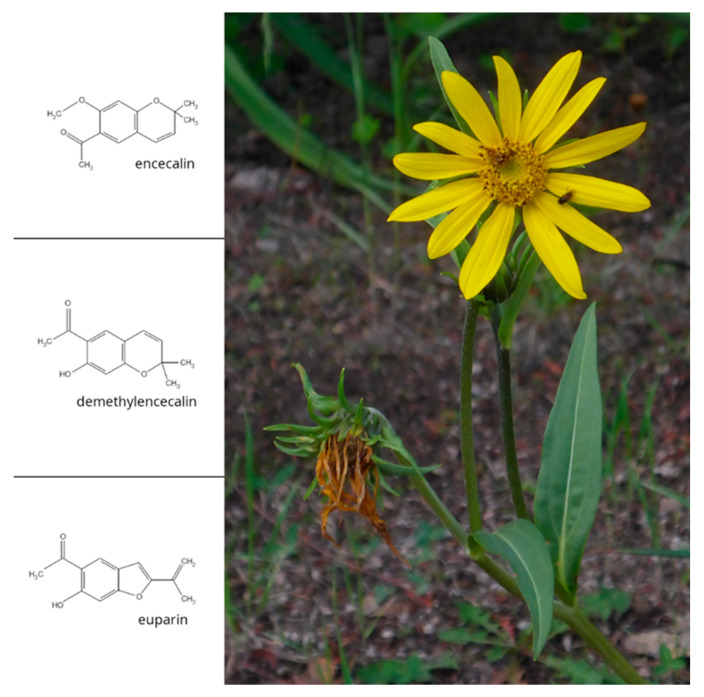
*Helianthella quinquenervis* and chemical structures of encecalin and related compounds demethylencecalin and euparin (photo of flowering plant: Bocoyna, Chihuahua; R. Bye).

**Figure 2 molecules-25-03231-f002:**
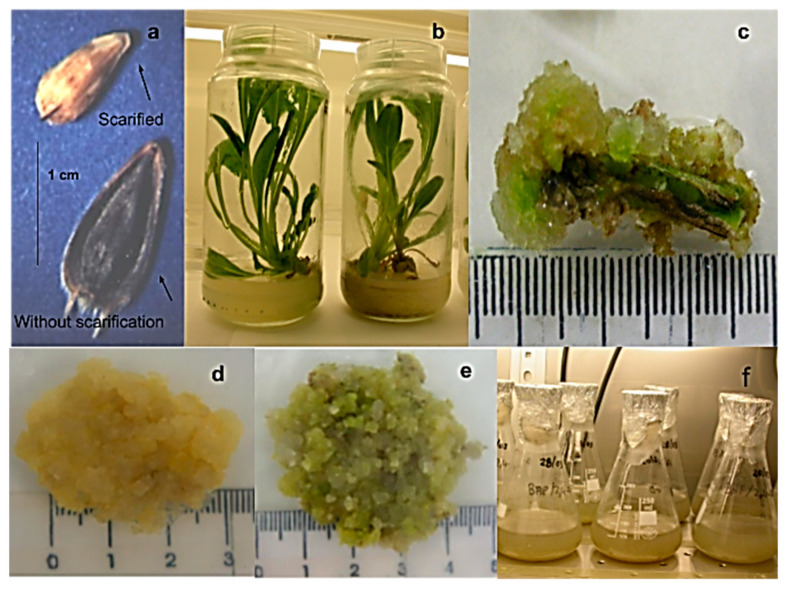
Seeds and cell cultures of *Helianthella quinquenervis*: (**a**) seeds with and without scarification; (**b**) two months in vitro plants; (**c**) leaf explant with forming callus; (**d**) callus in darkness; (**e**) callus in photoperiod; (**f**) cell suspension culture.

**Figure 3 molecules-25-03231-f003:**
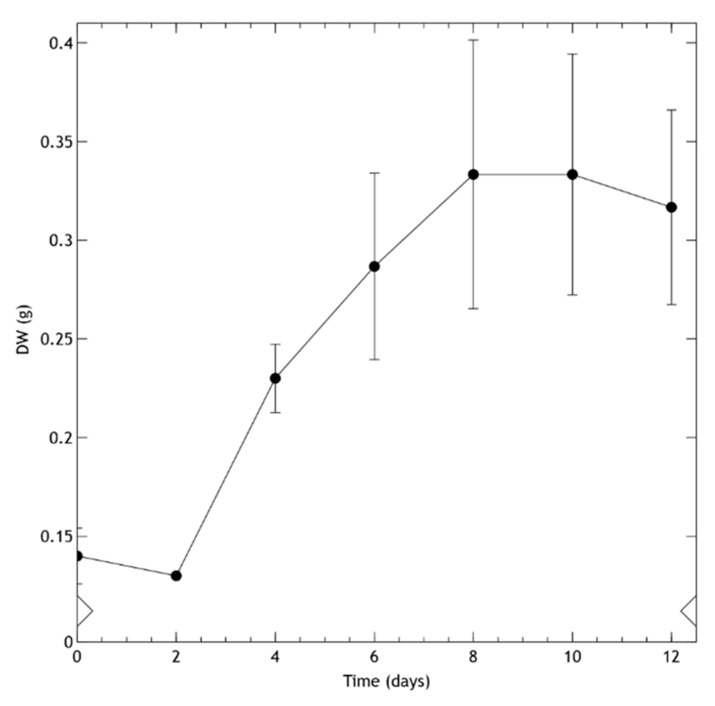
Kinetic growth of cell suspension culture of *Helianthella quinquenervis* expressed in dry weight (DW) g/150 mL. Values are means ± S.D. of the measurement from three flasks.

**Figure 4 molecules-25-03231-f004:**
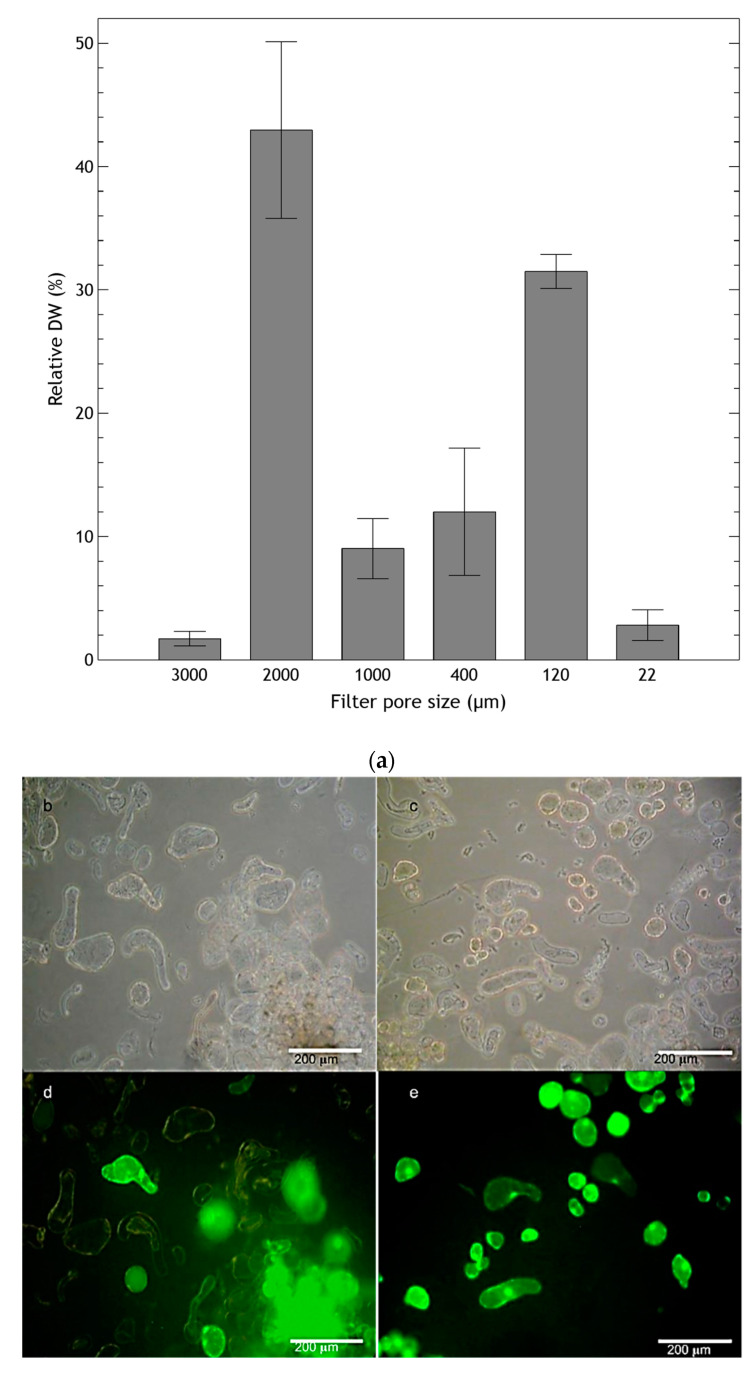
(**a**) Size of clusters of cell suspension culture of *Helianthella quinquenervis* analyzed at the stationary phase (eighth day). Results are expressed as the percentage of DW retained in each filter. Values are means ± S.D. of five measurements; (**b**–**e**) viability of two suspension culture fractions obtained by differential filtration with 2 mm mesh (**b**,**d**), and homogenous cultures obtained with 22 μm mesh (**c**,**e**) during the end of the exponential phase and the start of the stationary phase; (**b**,**c**) with light field; (**d**,**e**) with fluorescence microscopy.

**Figure 5 molecules-25-03231-f005:**
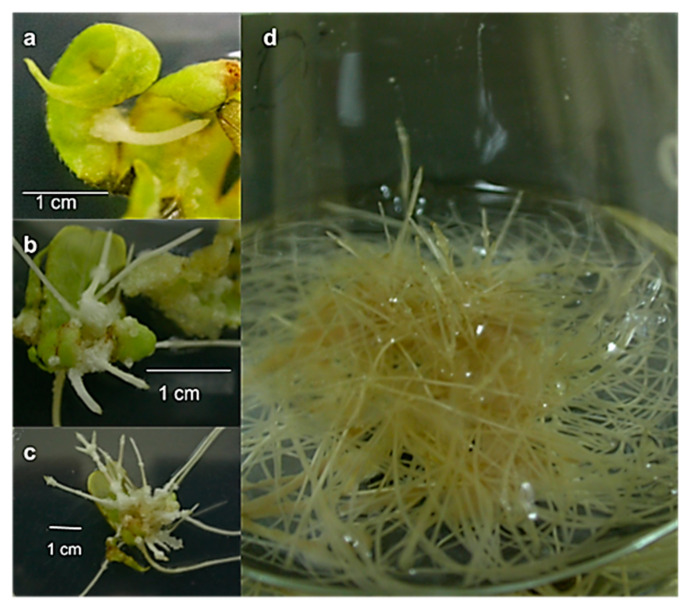
Hairy roots from *Helianthella quinquenervis* in hormone-free MS medium at (**a**) 12 d; (**b**) 20 d; (**c**) 30 d; (**d**) 90 d after transformation.

**Figure 6 molecules-25-03231-f006:**
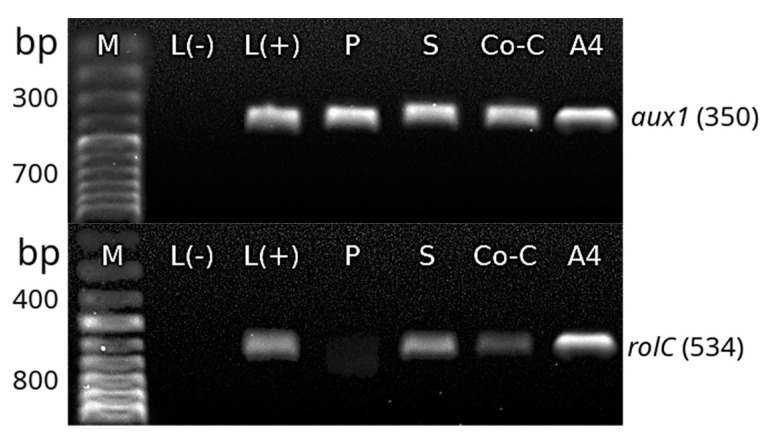
Polymerase chain reaction (PCR) analysis of *aux1* (350 bp), *rolC* (534 bp) and *virG* (350 bp) from T-DNA of the Ri of *A. rhizogenes*. Molecular marker (M); *Linum album* L(−) * (negative control); *Linum album* L(+)* (positive control); roots obtained by prick (P), sonication (S) and co-culture (Co-C); strain A4 of *A. rhizogenes* (A4); *H. quinquenervis* leaves (Hl) (negative control); wild roots of *H. quinquenervis* (negative control) (Hq). * Once nontransformed *H. quinquenervis* material was obtained, it was preferred as control.

**Figure 7 molecules-25-03231-f007:**
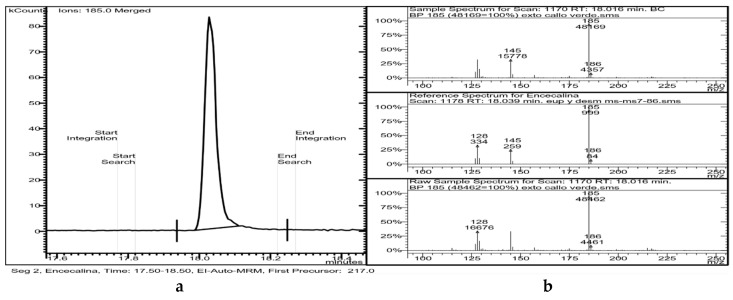
(**a**) Chromatogram of a sample of callus where the peak of encecalin can be seen, as well as (**b**) mass spectrum obtained by fragmentation of the precursor ion of mass 217.

**Figure 8 molecules-25-03231-f008:**
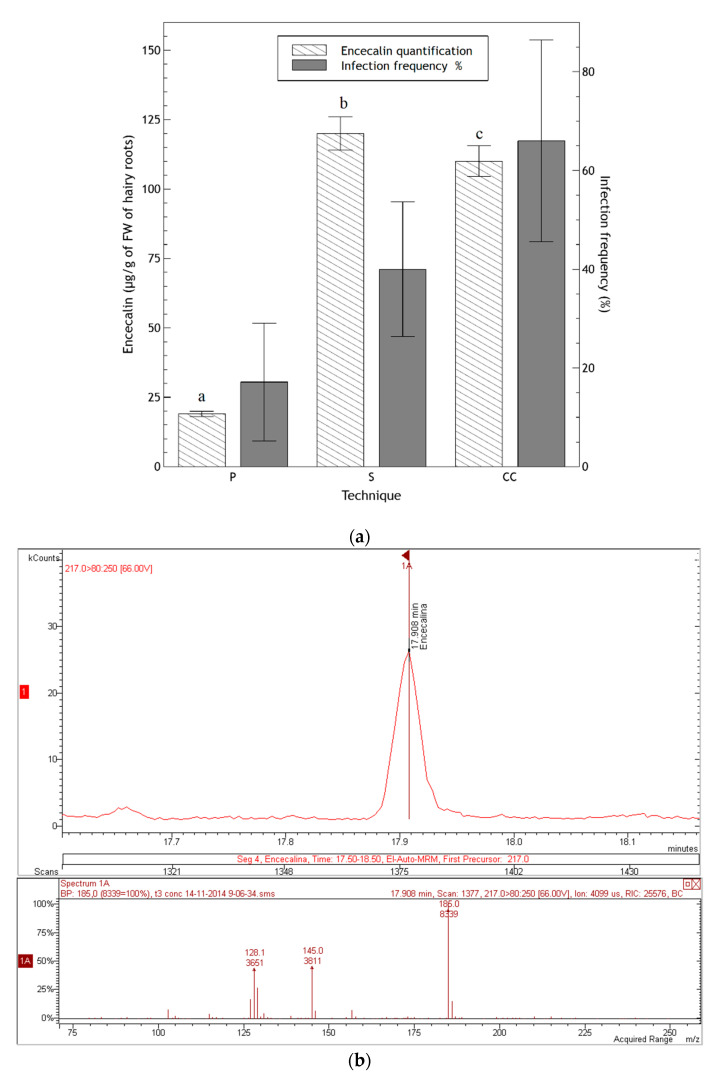
(**a**) Encecalin quantification in hairy roots (three months old) and percentage of transformation in three assays: prick (P), sonication (S) and co-culture (CC). Data of transformation frequency represent the average ± SD of five measurements. Data of encecalin quantification represent the average ± SD of three injections. Different letters indicate significant differences in encecalin production after a one-way ANOVA following by Tukey test (*p* < 0.05); (**b**) chromatogram of a sample of hairy roots (three months old) from co-cultivation where the peak of encecalin can be seen, as well as spectra of masses-masses obtained by fragmentation of the ion of mass 217.

**Figure 9 molecules-25-03231-f009:**
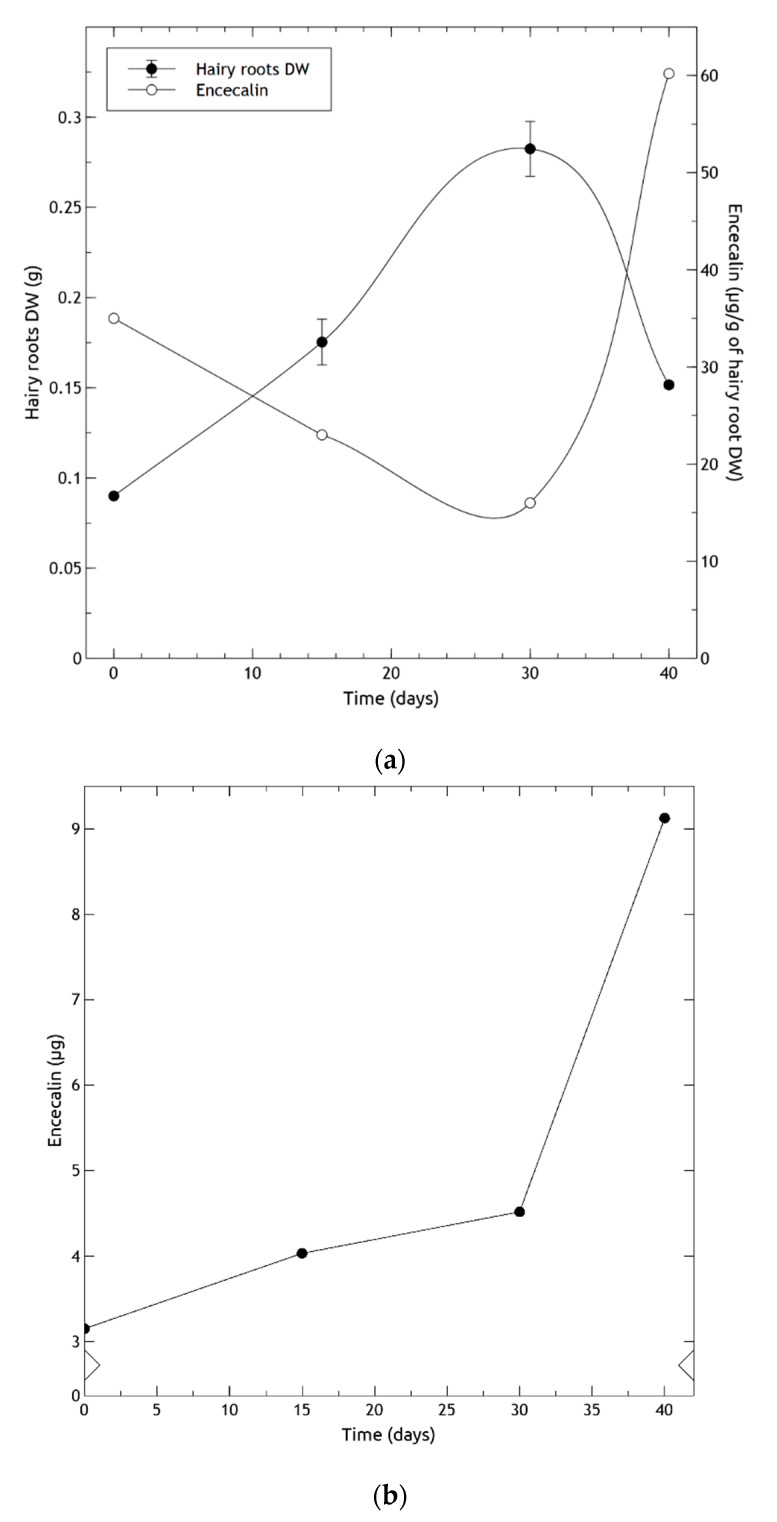
Encecalin production in six months old hairy roots of *Helianthella quinquenervis* obtained by co-culture; (**a**) Growth of hairy roots (•) and encecalin quantification (o); (**b**) encecalin production per explant obtained by multiplying the weight of explants (in grams) by their quantification per gram. DW data represent average ± SD of three measurements.

**Figure 10 molecules-25-03231-f010:**
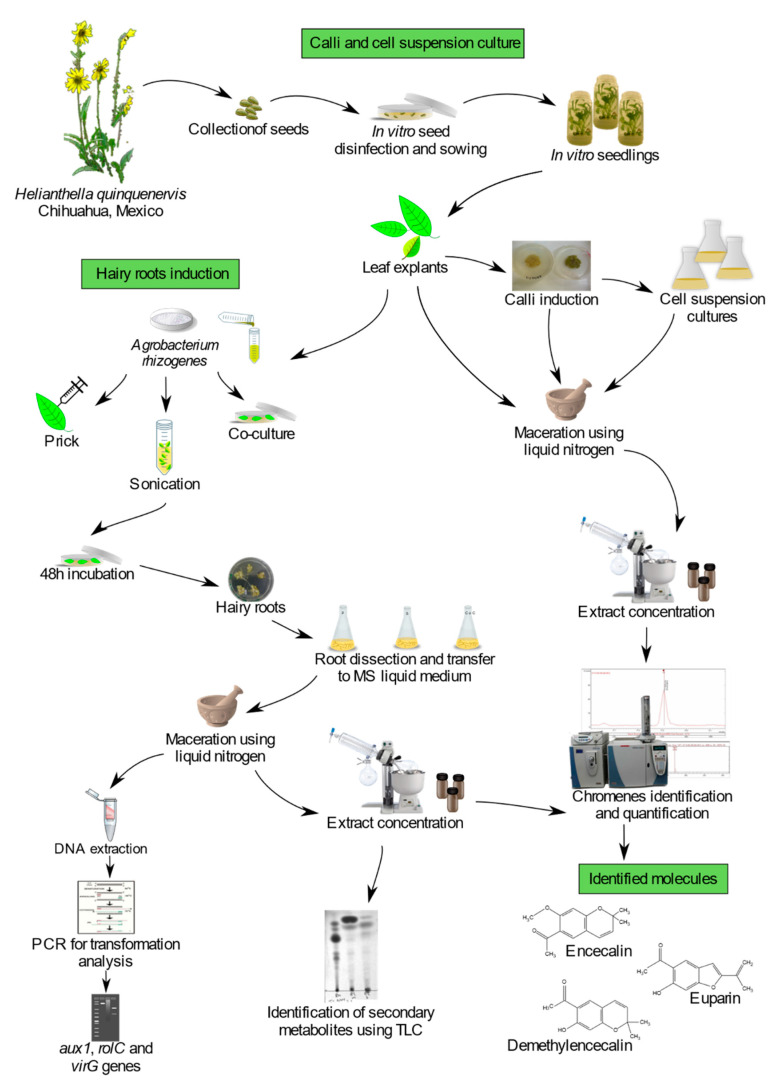
Experimental workflow.

**Table 1 molecules-25-03231-t001:** Hairy root formation on *Helianthella quinquenervis* leaf explants 40 days after transformation with *A. rhizogenes.*

Method of Transformation	Number of Explants Transformed/Total Explant	Range of Roots/Explant
Prick	6/35	1–3
Sonication	18/45	1–5
Co-culture	20/30	1–9

**Table 2 molecules-25-03231-t002:** Amount of encecalin quantified in *Helianthella quinquenervis* cultures.

Culture	Encecalin (µg/g FW)
Calli ^1^ (photoperiod)	165 ± 7
Calli (darkness)	290 ± 3
Cell suspension culture	327 ± 14
Hairy roots (co-culture)	1086 ± 31

^1^ Calli kept under a 16 h/light 8 h/darkness photoperiod. Hairy roots were obtained by the co-culture method. GC-MS-MS: gas chromatography with mass detection in masses-masses method. The cultures were kept for six months under the conditions described here before making the extractions.

**Table 3 molecules-25-03231-t003:** Specific pairs of primers used for PCR analysis of transformed roots of *Helianthella quinquenervis.*

Gene	Expected Amplified Product (bp)	Sequence
*rolC*	534	1: 5′-TAA CAT GGC TGA AGA CGA CC-3′ 2: 5′-AAA CTT GCA CTC GCC ATG CC-3′
*aux1*	350	1: 5′-TTC GAA GGA AGC TTG TCA GAA-3′ 2: 5′-CTT AAA TCC GTC TGA CCA TAG-3′
*virG* ^1^	350	1: 5′- ACT GAA TAT CAG GCA ACG CC-3′ 2: 5′- GCG TCA AAG AAA TAG CCA GC-3′

^1^ Sequence obtained from Takashi et al. [57].

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
