# Peer review of "Production of Encecalin in Cell Cultures and Hairy Roots of Helianthella quinquenervis (Hook.) A. Gray"

_molecules, 2020, doi:10.3390/molecules25143231_

Round 1

Reviewer 1 Report

The purpose of the study was to investigate the production of encecalin in cell cultures and hairy roots of Helianthella quinquenervis (Hook.) A. Gray. The presented results are significant, and the paper is quite well written, however, few imprevements are recommended:

  • Introduction is too long, and unnecessarily divided into so many paragraphs,
  • Figures 8-9 should be presented in a much higher resolution (description of results is illegible),
  • Student's test should be replaced with multifactorial ANOVA with subsequent post-hoc test (e.g. Tukey's test),
  • Discussion and interpretation of the results are superficial and should be thoroughly improved,
  • Gene names should be written in Italic, 
  • Extensive editing of English language and style required,
  • Experimental workflow (Figure 12) may be presented in more concise form.

Reviewer 2 Report

The manuscript describe the development of a method for the production of encecalin. For that, authors show different strategies for the optimization of the process and they finally remark the best conditions for encecalin production. The English level can be improved and the text can be clarified in some points, but in general it is easy to read. I have some comments that I hope will serve to improve the readability of the manuscript and some general questions.

Lines 130-147. The authors try different culture medium but, if I understand correctly, they discard both of them? (lines 138-139). However, after that, they mention the use of BAP + 2,4-D (line 147). Can you clarify that to me? I would suggest that you rewrite the parragraph. 

Figures can be combined (for example figures 4 and 5 or 9 and 10) as they mention the same point of the results. In general, the quality of the figures can be greatly improved.

Figure 7. Sizes in the marker are not indicated. 

Line 316 - decrease instead of increase

How do you define the stationary phase of the hairy roots growth?
